# Age and Sex Differences in the Use of Emergency Telephone Consultation Services in Saitama, Japan: A Population-Based Observational Study

**DOI:** 10.3390/ijerph17010185

**Published:** 2019-12-26

**Authors:** Akihisa Nakamura, Toshie Manabe, Hiroyuki Teraura, Kazuhiko Kotani

**Affiliations:** 1Chichibu City Otaki National Health Insurance Clinic, 925 Otaki, Chichibu City 369-1901, Saitama, Japan; m06066an@jichi.ac.jp; 2Division of Community and Family Medicine, Center of Community Medicine, Jichi Medical University, Shimotsuke-City 329-0498, Tochigi, Japan; manabe@jichi.ac.jp (T.M.); m05062ht@jichi.ac.jp (H.T.)

**Keywords:** ambulances, emergency medical services, family member, prehospital emergency care services, sex differences, telemedicine, telephone triage

## Abstract

The frequency of use of emergency telephone consultation (ETC) services, which is a telephone triage system in Japan, was explored to determine age- and sex-related trends on symptoms/events among Japanese adults. Data were obtained from records of the initial year of ETC services in Saitama Prefecture (from October 2014 to September 2015). Residents who used the ETC services were divided into four age groups (20–39, 40–64, 65–74, and ≥75 years old). The number of calls per 1000 persons (call rate, CR) was compared between the groups. The annual CR for the top 10 symptoms/events were assessed. The annual CR was 2.6/1000 persons. The annual CR was significantly higher for women (2.9) than for men (2.3) (*p* < 0.05). ETC use was the highest in the 20–39 age group for both sexes (3.3 and 4.4 in men and women, respectively). All groups had fever, abdominal pain, and nausea/vomiting in common. In older adults, the frequency of events such as bruises and head injury was high. Women and younger adults tended to use ETC services. There were similarities and differences in symptoms/events among the groups, which should be recognized by call centers to help foster call center staff.

## 1. Introduction

Emergency telephone consultation (ETC) services provide a telephone triage system for urgent care in Japan [1]. Similar telephone triage systems have been introduced in other developed countries [2,3,4,5,6,7,8,9,10]. In Japan, ETC services for adults were launched in Tokyo in 2007. A previous study on the appropriate use of ambulances reported that the ETC services in Tokyo contributed to a 3.7% decrease in the need for ambulance transport over the first year of operations [1]. Following these results, the Fire and Disaster Management Agency has promoted the national development of ETC services. The number of local governments introducing this service has gradually increased in Japan and approximately 40% of the population can now access the services [11]. In Saitama Prefecture, which has a population of 7.2 million, ETC services for adults were introduced in October 2014. However, the services operated only from 18:30 to 22:30.

Despite the growth of ETC services in Japan, little information has been published on the characteristics of adults’ use of ETC services. A previous study in Tokyo ETC services included all age groups, but did not analyze any age- or sex-related trends on the frequency of use of the services or on the symptoms/events. This knowledge about age- and sex-related trends may help to foster call center staff. This knowledge contributes to identify what symptoms/events for which generation should be focused on and trained for in call center staff.

The objectives of the present study were to show the frequency of the use of ETC services by age and sex using population data, and to determine age- and sex-related trends on the symptoms/events as the reason for adults’ use of the service.

## 2. Materials and Methods

### 2.1. Study Design and Data Collection

This population-based, observational study was performed in Saitama Prefecture, which is a large prefecture in Japan with a population of approximately 7.2 million. Data were obtained from Department of Public Health and Medical Services records in Saitama Prefecture for the first year of ETC services from October 2014 to September 2015. The recorded items were the general characteristics of users who used the ETC service, including date of use, place of residence, symptoms/events as the reasons to use ETC services, and the relationship of callers who made a call for users (oneself, family member, others or unknown). We defined users as residents who had symptoms/events and used ETC services, and callers as persons who actually called the ETC services for users. The data for users aged <20 years and users who did not record the necessary information including age, address or caller, and for those calling from outside of Saitama Prefecture were excluded.

First, we compared the annual call rate (CR, the number of ETC users per 1000 persons during an observational period) among men and women based on the results of the 2015 national census [12]. Second, the remaining users were divided into four age groups (20–39, 40–64, 65–74, and ≥75 years). Annual CRs were compared between groups by sex. In addition, the proportion of callers using ETC services was compared among user age groups. The annual number of calls and the annual CR for the 10 most frequent symptoms/events were assessed by age and sex.

This study was approved by the Jichi Medical University Research Ethics Committee (17-187). The Ethics Committee waived the need for informed consent due to the anonymity of the data collected for routine operations and the retrospective nature of this study.

### 2.2. ETC Services in Saitama Prefecture, Japan

In October 2014, the Saitama prefectural government launched ETC services for adults. These ETC services comprised nurse-led telephone services and operated only from 18:30 to 22:30. Eighty protocols were involved depending on the content of the telephone consultation. These protocols were developed based on examples of protocols for telephone consultation published by the Ministry of Health, Labor and Welfare of Japan. The users of ETC services in Saitama Prefecture were classified into the following six categories: (1) required ambulance transport, (2) visited a medical institution within approximately 1 h; (3) visited a medical institution within approximately 6 h; (4) non-urgent cases, visited a medical institution on the next day; (5) non-urgent cases, could be handled at home; and (6) other.

### 2.3. Statistical Analysis

The data were reported as percentages for categorical variables and medians and interquartile ranges for continuous variables. Annual CRs between men and women were compared using chi-squared tests. Annual CRs and the proportion of callers using ETC services were compared by age group using multiple comparisons adjusted by the Holm method.

The data were analyzed using EZR software (version 2.4-0; Saitama Medical Center, Jichi Medical University, Saitama, Japan) [13]. For all analyses, significance levels were two-tailed and *p* values < 0.05 were considered to indicate statistical significance.

## 3. Results

Overall, a total of 22,073 ETC users were recorded during the study period, 1586 of whom were excluded because of their young ages, i.e., <20 years. Figure 1 shows the outcomes of the ETC services. After excluding calls from outside of Saitama Prefecture and those with missing information, 15,257 calls (69.1%) were included in the analysis.

The total annual CR for ETC services in Saitama Prefecture was 2.6/1000 persons. The median age of ETC users was 43 years (interquartile range, 32–61 years). Table 1 shows the number of ETC users and the annual CR by sex and age. The annual CR of women (2.9/1000 persons) was significantly higher than that of men (2.3/1000 persons) (*p* < 0.05). For both men and women, ETC use was the highest in the 20–39 age group for both sexes (3.3 in men and 4.4 in women), and the lowest in the 65–74 age group (1.4 in men and 1.8 in women).

Table 2 shows the annual CRs for the 10 most frequent symptoms/events by age and sex. In each group, the 10 most frequent symptoms/events accounted for approximately 40%–50% of the total symptoms/events. All groups shared a common frequency of fever, abdominal pain and nausea/vomiting. Older adults were more likely than younger adults to have hypertension. Especially in older adults aged ≥75 years, the frequency of bruises and head injury was high. Regarding ETC use for older adults, the proportion of callers who were family members using ETC services was high (Figure 2), with the highest proportion (74%) among the group aged ≥75 years.

## 4. Discussion

During the study period, the total number of ETC calls was 15,257 and the total annual CR was 2.6 per 1000 persons in Saitama Prefecture, Japan. Women used ETC services more often than men. For both sexes, ETC use was highest among the 20–39 age group. The proportion of callers who were family members using ETC services was higher among older adults than among younger adults.

The annual CR in Saitama Prefecture was seemed to be low compared to that in previous studies in the UK [3] and Western Australia [4], where annual CRs in the initial year were reported. The annual CRs in the UK and Western Australia were 52 and 95 per 1000 persons, respectively. This may be partially because the data from the UK and Western Australia included children, while the subjects in this study were adults. The availability of ETC services may be another reason for this difference. ETC services were available in 24 hours a day in the UK and Western Australia while they were available only from 18:30 to 22:30 in Saitama Prefecture.

In terms of sex differences, the annual CR was significantly higher for women than for men. Three reports have explored the number of calls for ETC services by men and women in the telephone triage system. On the one hand, a previous study from Tokyo showed that the number of calls was slightly higher for men (13,801 calls) than for women (12,050 calls) [1]. On the other hand, two studies from US [7] and Canada [8] showed that the number of calls was higher for women than men (US: 8197 calls in men and 19,782 calls in women. Canada: 39,222 calls in men and 57,146 calls in women). These studies could not strictly compare the frequency of calls because the analysis was done without considering population differences between men and women. In the present study, the frequency of calls could be compared between men and women by measuring the annual CR while considering population by sex. The results indicated that women used ETC services more frequently than men did. This finding could be attributable to the fact that research on the social use of telephones has systematically shown a clear sex difference [14,15,16,17]: i.e., women use the telephone more often than men because of the social positions occupied by the sexes (women’s family role) and their psychological characteristics.

For both men and women, the use of ETC was the highest in the 20–39 age group for both sexes. The need for emergency medical services is increasing with the aging population in Japan [18]. In 2017, ambulances were dispatched 6.3 million times nationwide in Japan, which is a roughly 20% increase compared with 2007 [19]. The proportion of older adults (aged ≥ 65 years) using ambulance services increased from 47% to 59% between 2007 and 2017 [11]. In Japan, where the population is rapidly aging, supplying stable emergency medical services has become an increasingly urgent issue. The present study might suggest that ETC services were not frequently used by older adults, who were at high need for emergency medical services [20,21]. There are two possible hypotheses for explaining the result of the present study. The first is the different behavioral patterns of younger and older adults. Compared with older adults, younger adults have been found to be more likely to use a telephone in the course of their daily life [17,22]. The second is the difference between younger and older adults in terms of awareness of ETC services. Previous studies found that awareness of ETC services was greater among younger adults than among older adults [23,24]. This difference may be related to differences in the frequency of the use of ETC services between younger and older adults.

The 10 most frequent symptoms/events accounted for approximately 40%–50% of the total symptoms/events. One point that younger and older adults have in common concerning their symptoms was a high incidence of fever, abdominal pain and nausea/vomiting. A study conducted in Australia and New Zealand, where symptoms are reported in detail, also found a high incidence of fever, abdominal pain and nausea/vomiting, similar to the results of the present study [25]. While hypertension was not included in the 10 most frequent symptoms/events in adults aged ≤64 years, it was included in adults aged ≥65 years. This was attributed to the increase in the prevalence of hypertension that occurs with aging [26]. Symptoms related to falls, such as bruises and head injury, were particularly common in adults aged ≥75 years. This could be explained by the fact that the risk of falls was increasing with aging [18].

Among adults aged ≥65 years, ETC use was more common among those aged ≥75 years than among those aged 65–74 years. One possible reason for this finding is that, compared with those aged 65–74 years, a greater proportion of those aged ≥75 years may have had family members acting on their behalf. The fact that the use of ETC services among older adults was prompted by calls from family members suggests that they are possibly a key factor in the use of ETC services by older adults.

According to a 2014 government white paper on information and communications in Japan, the rate of Internet use among individuals aged 20–59 years is more than 91% [27]. The rate of e-health literacy (i.e., the ability to appropriately retrieve, assess and utilize health information on the Internet to utilize such information effectively) was reported to be particularly high among those aged in their 40s and 50s [28]. Consequently, using the Internet to disseminate information about ETC services may be an effective means of encouraging its use by younger and middle-aged adults.

This study has several strengths. First, this study provides important information about the frequency of use of ETC services for each generation of the broader population. Second, this was a population-based study conducted in a large, populous prefecture with a population of 7.2 million. Third, this is one of the few studies that focus on callers, which suggests that family members are possibly a key factor to encourage older adults to use ETC services.

This study has several limitations. First, ETC services in Saitama Prefecture have a limited operation time, which may affect the frequency of symptoms/events. Therefore, the findings of this study may not be generalizable to ETC services operating under different times and conditions. Second, only approximately 70% of users were included in the analysis because of the lack of necessary information including age, sex, and address, or the identity of the caller. Therefore, the actual frequency of use of ETC services may be higher than that reported in this study. Third, this study suggests that family members are possibly a key factor to promote older adults’ use of ETC services. However, we were unable to refer to the characteristics of family members, e.g., what kind of relationship between the ETC service user and family member, and whether the family member lives together with the service user. Finally, we were unable to identify persons who used ETC services more than once during the study period. To calculate the CR, we used only the number of ETC calls by sex and age. Therefore, the same persons might have made multiple ETC calls, which would have led to a lower actual CR than that reported in the present study.

## 5. Conclusions

The frequency of use of ETC services by age and sex using population data and age- and sex-related trends on the symptoms/events was reported. Women and younger adults tended to use ETC services. There were some common symptoms/events to both men and women at any age, while there were other symptoms/events that depended on age and sex. When more local governments introduce ETC services in their municipalities in the future, it is necessary to think about the methods of ETC services that are easy to use for all ages and genders. It also crucial to provide the epidemiological information to call center staff to enhance their services. This topic warrants further additional study.

## Figures and Tables

**Figure 1 ijerph-17-00185-f001:**
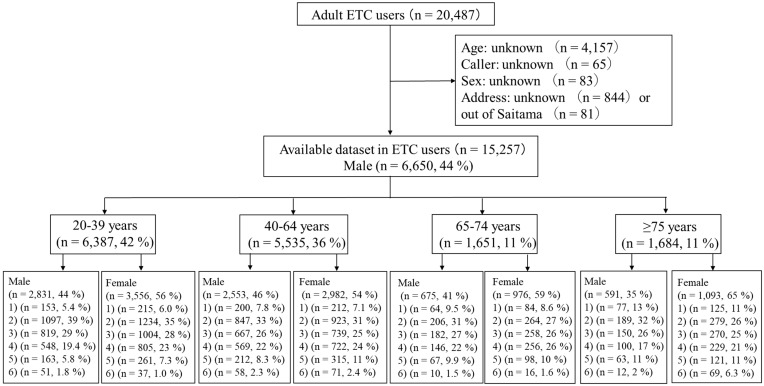
Outcome of emergency telephone consultation (ETC) services in Saitama, Japan. The users of ETC services in Saitama Prefecture were classified into the following six categories: (1) required ambulance transport; (2) visited a medical institution within approximately 1 h; (3) visited a medical institution within approximately 6 h; (4) non-urgent case, visited a medical institution on the next day; (5) non-urgent case, could be handled at home; and (6) other.

**Figure 2 ijerph-17-00185-f002:**
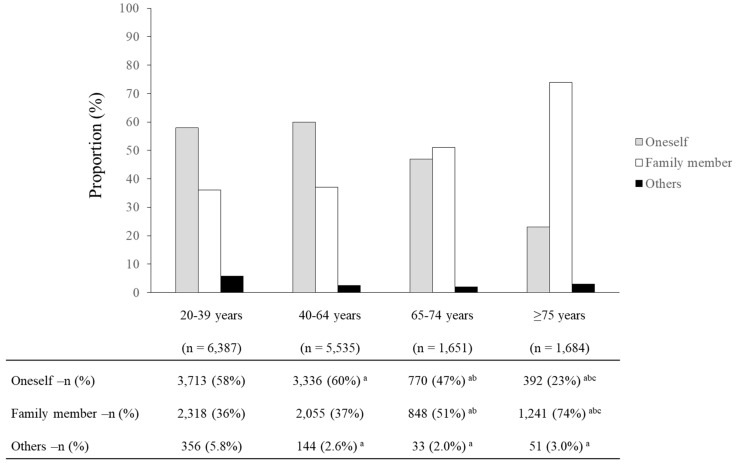
Proportion of caller by age on using ETC services in Saitama, Japan. ETC, emergency telephone consultation. ^a^ vs. 20–39 years: *p* < 0.05; ^b^ vs. 40–64 years: *p* < 0.05; ^c^ vs. 65–74 years: *p* < 0.05. Multiple comparisons adjusted by the Holm method.

**Table 1 ijerph-17-00185-t001:** Annual call rate (CR) by sex and age using multiple comparison.

Sex	Total			20–39 Years		40–64 Years		65–74 Years		≥75 Years	
	ETC Users/Population	Annual CR	*p*-Value	ETC Users/Population	Annual CR	ETC Users/Population	Annual CR	ETC Users/Population	Annual CR	ETC Users/Population	Annual CR
Men	6650/2,943,898	2.3	<0.001 *	2831/859,561	3.3	2553/1,271,521	2.0 ^a^	675/491,988	1.4 ^a,b^	591/320,828	1.8 ^a,c^
Women	8607/2,999,236	2.9		3556/809,714	4.4	2982/1,213,603	2.5 ^a^	976/530,622	1.8 ^a,b^	1,093/445,297	2.5 ^a,c^

* chi-squared test; ^a^ vs. 20–39 years: *p* < 0.05; ^b^ vs. 40–64 years: *p* < 0.05; ^c^ vs. 65–74 years: *p* < 0.05. CR, call rate; ETC, emergency telephone consultation; Annual CR, the number of ETC calls per 1000 persons; Multiple comparison adjusted by the Holm method.

**Table 2 ijerph-17-00185-t002:** Annual CR for the 10 most frequent symptoms/events by age and sex.

		Overall		Men		Women	
	Rank	Symptom/Event (*n*, %)	Annual CR	Symptom/Event (*n*, %)	Annual CR	Symptom/Event (*n*, %)	Annual CR
20–39 years (*n* = 1,669,275)	1	Fever (1107, 17.3%)	0.66	Fever (527, 18.6%)	0.61	Fever (580, 16.3%)	0.72
	2	Abdominal pain (555, 8.7%)	0.33	Abdominal pain (205, 7.2%)	0.24	Abdominal pain (350, 9.8%)	0.43
	3	Nausea/vomiting (330, 5.2%)	0.20	Headache (123, 4.3%)	0.14	Nausea/vomiting (236, 6.6%)	0.29
	4	Headache (313, 4.9%)	0.19	Laceration (106, 3.7%)	0.12	Headache (190, 5.3%)	0.23
	5	Rash/hives (206, 3.2%)	0.12	Cold (95, 3.4%)	0.11	Rash/hives (124, 3.5%)	0.15
	6	Cold (203, 3.2%)	0.12	Nausea/vomiting (94, 3.3%)	0.11	Cold (108, 3.0%)	0.13
	7	Laceration (188, 2.9%)	0.11	Upper limb problem (90, 3.2%)	0.10	Diarrhea (87, 2.4%)	0.11
	8	Upper limb problem (170, 2.6%)	0.10	Contusion (83, 2.9%)	0.10	Ophthalmology-related (86, 2.4%)	0.11
	9	Diarrhea (164, 2.6%)	0.10	Diarrhea (77, 2.7%)	0.09	Laceration (8.2, 2.3%)	0.10
	10	Ophthalmology-related (163, 2.6%)	0.10	Ophthalmology-related (77, 2.7%)	0.09	Upper limb problem (80, 2.2%)	0.10
40–64 years (*n* = 2,485,124)	1	Fever (516, 9.3%)	0.21	Fever (261, 10.2%)	0.21	Fever (255, 8.6%)	0.21
	2	Abdominal pain (393, 7.1%)	0.16	Abdominal pain (193, 7.6%)	0.15	Abdominal pain (200, 6.7%)	0.16
	3	Headache (235, 4.3%)	0.09	Ophthalmology-related (105, 4.1%)	0.08	Headache (158, 5.3%)	0.13
	4	Ophthalmology-related (203, 3.7%)	0.08	Laceration (93, 3.6%)	0.07	Nausea/vomiting (125, 4.2%)	0.10
	5	Nausea/vomiting (186, 3.7%)	0.07	Upper limb problem (86, 3.7%)	0.07	Ophthalmology-related (98, 3.3%)	0.08
	6	Laceration (173, 3.1%)	0.07	Headache (77, 3.0%)	0.06	Dizziness (88, 3.0%)	0.07
	7	Upper limb problem (165, 3.0%)	0.07	Numbness (76, 3.0%)	0.06	Rash/hives (81, 2.7%)	0.07
	8	Rash/hives (150, 2.7%)	0.06	Rash/hives (69, 2.7%)	0.05	Laceration (80, 2.7%)	0.07
	9	Numbness (147, 2.7%)	0.06	Contusion (69, 2.7%)	0.05	Upper limb problem (79, 2.6%)	0.06
	10	Dizziness (146, 2.6%)	0.06	Problems from ankle to toe (65, 2.5%)	0.05	Chest pain (74, 2.5%)	0.06
65–74 years (*n* = 1,022,610)	1	Abdominal pain (107, 6.5%)	0.10	Fever (53, 7.9%)	0.11	Abdominal pain (69, 7.1%)	0.13
	2	Fever (105, 6.4%)	0.10	Abdominal pain (38, 5.6%)	0.08	Hypertension (55, 5.6%)	0.10
	3	Hypertension (77, 4.7%)	0.08	Nausea/vomiting (29, 4.3%)	0.06	Fever (52, 5.3%)	0.10
	4	Nausea/vomiting (75, 4.5%)	0.07	Dizziness (29, 4.3%)	0.06	Nausea/vomiting (46, 4.7%)	0.09
	5	Dizziness (75, 4.5%)	0.07	Numbness (25, 3.7%)	0.05	Dizziness (46, 4.7%)	0.09
	6	Ophthalmology-related (62, 3.8%)	0.06	Consciousness disorder (22, 3.3%)	0.04	Ophthalmology-related (41, 4.2%)	0.08
	7	Bite (57, 3.5%)	0.06	Hypertension (22, 3.3%)	0.04	Bite (38, 3.9%)	0.07
	8	Contusion (50, 3.0%)	0.05	Upper limb problem (22, 3.3%)	0.04	Contusion (35, 3.6%)	0.07
	9	Numbness (48, 2.9)	0.05	Ophthalmology-related (21, 3.3%)	0.04	Headache (33, 3.4%)	0.06
	10	Rash/hives (46, 2.9%)	0.04	Nose problem (19, 2.8%)	0.04	Rash/hives (30, 3.0%)	0.06
≥75 years (*n* = 766,125)	1	Fever (150, 8.9%)	0.20	Fever (67, 11.3%)	0.21	Fever (83, 7.6%)	0.19
	2	Nausea/vomiting (81, 4.8%)	0.11	Nausea/vomiting (38, 6.4%)	0.12	Contusion (50, 4.6%)	0.11
	3	Hypertension (68, 4.0%)	0.09	Consciousness disorder (27, 4.6%)	0.08	Hypertension (47, 4.3%)	0.11
	4	Contusion (65, 3.9%)	0.08	Abdominal pain (24, 4.1%)	0.07	Nausea/vomiting (43, 3.9%)	0.11
	5	Abdominal pain (60, 3.6%)	0.08	Hypertension (21, 3.6%)	0.07	Head injury (41, 3.8%)	0.09
	6	Consciousness disorder (58, 3.4%)	0.08	Numbness (20, 3.4%)	0.06	Dizziness (38, 3.5%)	0.08
	7	Dizziness (57, 3.4%)	0.07	Dizziness (19, 3.2%)	0.06	Abdominal pain (36, 3.3%)	0.08
	8	Head injury (53, 3.1%)	0.07	Nose problem (18, 3.0%)	0.06	Ophthalmology-related (35, 3.2%)	0.07
	9	Leg problem (51, 3.0%)	0.07	Leg problem (16, 2.7%)	0.05	Leg problem (35, 3.2%)	0.07
	10	Ophthalmology-related (46, 2.7%)	0.06	Contusion (15, 2.5%)	0.05	Facial and extremity injury (33, 3.0%)	0.07

Annual CR, the number of emergency telephone consultation calls per 1000 persons.

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
