# Peer review of "Age and Sex Differences in the Use of Emergency Telephone Consultation Services in Saitama, Japan: A Population-Based Observational Study"

_ijerph, 2019, doi:10.3390/ijerph17010185_

Round 1

Reviewer 1 Report

The paper is interesting and presents a useful insight into the healthcare facilities provided in the framework of telemedicine and similar fields in Japan. Although poorly generalizable to other countries, since the technological readiness in Japan is already much higher than in most Western countries, the results are quite interesting and could drive to an optimized organization of healthcare systems, with a significant cost saving without affecting the quality of the service.

I do not have particular concerns about the validity of the proposed approach or about the paper itself. I just would add some more practical implications of the research conducted in the re-organization of the healthcare assistance at a local and, eventually, national level.

Overall, I would suggest to carefully revising English language and grammar, as it should be improved throughout the text (probably a native speaker for revision could help).

Reviewer 2 Report

Thank you for submitting this well written manuscript for publication consideration. This is a large population study of emergency call use. The sample and literature review appear to be robust. The methodology for data analysis is sound and the data analysis and comparison with other research in this area is well done. My main recommendation would be to speak a bit more to why this study is important. A statement is made that this information is important to "help foster call centre staff" I am not entirely clear on what this means. Will the study findings, for example, impact the training of staff or how staff respond to calls? What are the implications of this research for practice, administration, education or further research?

Reviewer 3 Report

This is a very interesting study into the use of emergency calling centres for consultation in Japan.

The findings are also interesting, but they are based on descriptive statistics and univariate analysis only, whereas it would be interesting to see whether the significant relationships could be further explored in combination.

For example the  interpretation of the findings regarding the difference between men and women in the frequency of using the emergency call services could be explored further by considering the relationship between reason for calling he service, by whom the call is made for and by sex.

Also the explanations provided on women using the telephone more often than men because of the social positions occupied by the sexes (women’s family role) and their psychological characteristics needs to be clarified further.

Moreover, do the authors have information on the type of phone the call was made from i.e. domestic or mobile, as this could partly explain the high use among the 20-39 age group. Is there a charge to use the service and if so is there a difference depending on the type of device used?

Finally, it would be a good idea to suggest a comparison with the use of the ETC in other regions of Japan to detect any regional differences.

Reviewer 4 Report

It's interesting to study the use of emergency telephone consultation services in a rapidly aging nation as Japan. Results obtained could be further elaborated.
